# Some Interventions to Shift Meta-Norms Are Effective for Changing Behaviors in Low- and Middle-Income Countries: A Rapid Systematic Review

**DOI:** 10.3390/ijerph19127312

**Published:** 2022-06-14

**Authors:** Annette N. Brown

**Affiliations:** FHI 360, Washington, DC 20009, USA; abrown@fhi360.org

**Keywords:** social norms, meta-norms, behavior change, systematic review, rapid review, gender, program design, health behaviors, intimate partner violence, HIV

## Abstract

Social-norms approaches are increasingly included in behavior-change programming. Recent reviews categorize a large number of norms-shifting programs but do not synthesize evidence about effectiveness. To inform the design of social and behavior-change programs in low- and middle-income countries in response to time-sensitive demands, this rapid systematic review examines the evidence for the effectiveness of interventions that use norms-based approaches to change behavior. Nine indexes and eight websites were electronically searched for both systematic reviews and primary studies. Abstracts and full texts were screened to include: documents published in 2010 and later; documents evaluating the effectiveness of programs that include norms-based approaches; documents measuring behavioral outcomes; and documents employing quantitative analysis of concurrent treatment and comparison groups. Data collected include participant age cohort, program name and duration, scope of norms, intervention activities, category of behavioral outcome, and statement of findings for the main behavioral outcome(s). Primary studies were appraised based on identification strategy. Search and screening yielded 7 systematic reviews and 29 primary studies covering 28 programs. Across the primary studies, the programs are highly heterogeneous, and the findings are mixed, with some strong positive effects and many marginal or null effects on behavior change. Taken together, the evidence shows that meta-norms-based approaches can be part of effective programs but do not assure that programs will change behaviors. Program designers can draw some general conclusions from this review but can also use it to locate specific studies relevant to their evidence needs.

## 1. Introduction

Social-norms approaches are increasingly included in behavior-change programming [1]. The definition of social norms, simply, is what other people do, believe, and approve of. Behavior-change interventions based on the theory of social norms, ascribed to Perkins and Berkowitz (1986) [2], seek to influence behaviors by providing feedback about existing social norms to change individuals’ perceptions of social norms [2]. The original example in health is binge-drinking behavior in colleges, where showing students that binge drinking is actually less prevalent than they perceived led to decreased binge drinking [2]. Dempsey, et al. (2018) review the evidence for the social norms theory approach for health attitudes and behavior change [3]. Social norms theory is also used to design programs in sectors other than health; a well-known example is social norms information about energy usage reported in letters to households that led to changes in energy conservation [4].

Social-norms approaches also include norms-shifting interventions. These interventions seek to do more than simply provide feedback about current norms. They seek to change norms, particularly norms in the sense of what other people approve of. For example, the prevailing norm may be that the perpetration of intimate partner violence (IPV) is socially accepted. A norms-shifting intervention attempts to change that norm so that individuals increasingly perceive that others do not approve of that behavior. A number of recent articles and reports usefully review theories, define concepts, and describe approaches for social norms interventions to change behavior [5,6,7]. Labels differ across articles, with some authors using norms-shifting or norms-changing to mean all interventions with a norms approach. Heise and Manji (2016) [1] define meta-norms as norms that influence multiple behaviors. They give the example of the norm that violence is not an appropriate form of punishment. Shifting toward that norm may reduce corporal punishment in schools and abuse within households. The Social Norms Lexicon similarly defines meta-norms as foundational norms that serve to create and maintain social norms and, as such, may be harder to shift [8].

A few recent reviews catalog and categorize a large number of norms-shifting programs [9,10,11] but do not report, assess, or synthesize evidence for the effectiveness of these programs, especially not evidence from rigorous evaluations. Thus, it is difficult for decision makers to access effectiveness evidence for program design.

This present review was conducted to meet the needs of my own organization, at which program designers need to respond to opportunities with fixed deadlines. Often, practitioners rely on limited or weak evidence when writing proposals—evidence from best practices based on past programming, one or a handful of academic articles, non-rigorous program evaluations, and even anecdotes. This review is the second pilot review in an initiative to select and refine rapid-review methodologies to provide higher-quality evidence for time-sensitive design work and to share that evidence publicly [12,13,14]. Originally planned for six weeks, the review ultimately took ten weeks from protocol registration to draft manuscript, chiefly due to staffing challenges, but I was able to share initial findings with colleagues as I was conducting the review.

### 1.1. Rapid Systematic Reviews

Although there is general agreement that rapid-review methodologies produce less reliable findings than full systematic reviews, there is growing demand by decision makers for timely evidence reviews [15,16]. One systematic review of rapid-review studies and their methods finds heterogeneity across methods and ambiguous definitions and standards, including a range of timelines from 1 to 12 months [16]. Another systematic review of rapid reviews finds 82 reviews that use a variety of streamlined methods [17]. An experiment comparing three full systematic reviews to three rapid reviews each for the same questions concludes that rapid reviews “may be feasible for focused clinical questions” (p. 9) but is more skeptical about the sensitivity of a rapid review to a systematic review in the case of complex programs [15]. Nevertheless, rapid reviews can serve the needs of decision makers [18], and to help meet those needs, the World Health Organization has published a guide for rapid reviews in public health [19].

Some rapid systematic reviews limit their searches to existing systematic reviews, some to primary studies, and some include both. The full rapid review conducted here includes both systematic reviews and primary studies as eligible studies. If there are relevant systematic reviews, they may provide program-ready findings about what works, so it is important to seek them out. The systematic reviews found for this review do not provide such findings. In the interest of space, I briefly summarize the findings from the reviews in this article and present the methods, tables, and summaries in the Appendix A.

### 1.2. Objectives

To inform the design of social and behavior-change programs in low- and middle-income countries in response to time-sensitive opportunities, this rapid systematic review addresses the research question stated in the pre-registered protocol, “What is the recent evidence about the effectiveness of behavior-change interventions that incorporate social-norms approaches in improving behavioral outcomes in support of improving lives in low- and middle-income countries?” Specifically, this manuscript covers studies of interventions that address meta-norms, which are norms intended to influence multiple behaviors. The review reports the scope of the evidence base and provides information about studies to allow program designers to find evidence relevant to their needs. It also explores whether there are consistent features across effective programs.

## 2. Materials and Methods

The protocol for this review was registered on Open Science Framework on 30 October 2020 (osf.io/3fpnq) in a document modeled on the PROSPERO template [20]. The PRISMA (Preferred Reporting Items for Systematic Reviews and Meta-Analyses [21]) statement checklist is Appendix A. The following features of the review design were selected to enable the systematic review to be rapid. The range of eligible publication dates was limited to 2010 to present. This range allowed the review to capture the most recent primary studies and incorporate earlier research through systematic reviews. The electronic search strategies required that the word “norms” appear in the title, abstract, or keywords, and other terms in the string were tested and selected to generate a number of hits that was reasonably large while still feasible to screen within the timeline. The search process did not include a manual search of reference lists or expert consultation. A single reviewer conducted all screening and coding.

To avoid duplication, I conducted a quick search during protocol development for published systematic reviews matching the review objectives and found none. Systematic review protocols were not searched, as there was no guarantee that these reviews would be completed during the required timeline.

Additional details regarding materials and methods are presented in Appendix B.

### 2.1. Eligibility Criteria

The inclusion criteria for primary studies, listed roughly in order of ease of exclusion, were: (1) published in 2010 or later; (2) published in English; (3) journal article, published institutional report, published institutional working paper, or publicly posted pre-print; (4) reports findings for at least one endline; (5) evaluates an intervention implemented in a low- or middle-income country; (6) evaluates an intervention implemented in the field (i.e., excludes laboratory experiments and lab-in-the-field games); (7) measures a behavioral outcome; (8) uses quantitative study design with data from concurrent intervention and comparison groups using random assignment, natural experiment, fixed effects, or multivariate analysis identification strategies; (9) evaluates an intervention designed or intended to change behavior and includes a norms-based approach; (10) full text is available through our library without excessive cost.

The restriction to English-language publications facilitated the rapid review and accommodated the skills of the reviewers; however, it also ensures that the included studies are accessible to our colleagues using the evidence. This restriction also limits the review as discussion under limitations. All documents passing title and abstract screening were available in full text and screened, so the availability restriction did not bind. For inclusion, behavioral outcomes could be self-reported but needed to measure the actual behavior of the respondent. This definition rules out measures of behavioral intent and measures of the experience from others’ behavior. This latter clarification affected the inclusion of intimate partner and gender-based violence studies, as some of these studies measure whether the respondent experienced violence but do not measure the respondent’s behavior.

The most subjective of the inclusion criteria was the determination that there was a norms-based approach for behavior change. Although the term “norms” was used in the search strings, not all of the studies described the evaluated interventions as norms-based or social norms programs. I applied the eligibility criterion by examining the programs’ activities and the stated objectives of those activities. I looked for the use of actual norms, which are the attitudes held and behaviors exhibited by other people within a social group, or subjective norms, which are one’s perceptions of what most other people within a social group believe and do. I considered norms-based approaches for behavior change to include: activities to change the attitudes of a social group with the intention that the changed attitudes constitute new norms that influence individuals’ behavior; activities to change the behaviors of a social group with the intention that the changed group behavior establishes new norms that influence individuals’ behavior; and activities to inform or change one’s perception of what a social group believes or does with the intention of influencing that person’s behavior.

Note that activities to change group, or community, behavior without the stated intention to shift norms were not counted as norms-based approaches here. For example, a community sanitation program that seeks to increase sanitation behaviors across the community is not a norms-based intervention simply because it is community-based. However, such a program that includes activities to make improved behavior visible with the intent to influence others—for example, participants receive public acknowledgement for installing a toilet—does count as having a norms-based approach.

The word “norms” is sometimes also used to mean methods or processes. For example, Blattman, Hartman and Blair (2014) [22] study an intervention to train communities in the “practices and norms” of alternative dispute resolution. Here, “norms” means how people generally do something rather than what a social group believes or does. “Norms-teaching” interventions, which can be thought of as how-to rather than how-should interventions, were not included in this review. 

### 2.2. Search Strategy

To better capture a wide variety of programs with norms-based approaches, the search strategy used a simple set of thematic terms—focusing on norms and behavior change—instead of attempting to name all the possible intervention types. The strategies also included a set of terms for interventions and a set of terms for evaluation. Figure 1 shows the search strategy for EBSCO minus the separate country terms for all low- and middle-income countries.

To reduce publication bias and to reduce “publication lag” bias (caused by the longer lag between endline data collection and journal publication for social science impact evaluations than for health impact evaluations [23]), we conducted a website search of eight websites that provide databases of systematic reviews and of impact evaluations for low- and middle-income countries. The librarian conducted the electronic searches on 2 November 2020, and the second reviewer conducted the website searches between 2 and 6 November 2020. Appendix A catalogues the complete search strings and the number of hits for the index and website searches.

### 2.3. Selection

After screening, to focus this first manuscript from the review on the types of programs most relevant to my organization’s current design needs, I coded the included studies for whether the evaluated norms-based approach employs direct norms or meta-norms [1]. An example of a direct norm is the prevalence of condom use in a peer group, which is used to influence condom-use behavior. Examples of meta-norms are gender rights and HIV risk prevention (Although HIV risk prevention is specific to one disease, programs promote this norm in order to influence multiple behaviors, such as condom use, testing, and partner concurrency). Most authors do not make the distinction between direct norm and meta-norm when describing their programs, so I made the judgment based on the descriptions, not on labels used in the studies. This manuscript reviews the evidence for the studies of programs employing meta-norms.

### 2.4. Data Collection

For the primary studies, I coded a primary sector for each study and a second sector for those that spanned more than one. I coded the participants by predominant cohort—children (roughly 0–12), adolescents (roughly 13–18), youth (roughly 16–24), and adults. I also coded whether the primary behavioral outcome(s) was measured for males and females together, both males and females separately, only females, or only males. I extracted the program name and the duration of the intervention in months, defined by when program activities began and ended. An intervention of any duration less than one month was coded as one. In the overview table, I briefly describe the types of norms addressed by each program. The key findings tables include brief descriptions of the activities included in the interventions.

To categorize the behavioral outcomes, I generalized the six domains laid out by Shelton [24] for health systems. The six categories for the review are personal, help seeking, continuation, provider (or employee), pro-social (or anti-social), and decision making. As Shelton explains, there is overlap between some of these categories, so the coding involved some judgment calls. I coded outcomes related to the perpetration of IPV as continuation behaviors, as IPV happens as a repeated cycle of violence in intimate partner relationships. I coded other gender-based violence outcomes as personal. For findings, I extracted the document’s statement of the effect on the main behavioral outcome(s) after confirming these statements against the estimates and statistics provided in the tables. In some studies, the behavioral outcomes are reported as secondary outcomes.

Studies were coded as including cost-effectiveness analysis, including some cost information, or not including any cost information. The extracted sample size is for the sample used to estimate the behavioral effect. 

### 2.5. Appraisal

For the primary studies, given the need for rapid appraisal and to enable the participation of a junior reviewer on the team, as originally planned, the appraisal methodology was largely tied to the identification strategy. The first step in appraisal was to determine the identification strategy for the effect estimation for the primary behavioral outcome. I used the hierarchy from Waddington, et al.’s recommendation for risk-of-bias assessment, which has four levels: randomized controlled trial, natural experiment (i.e., as-if random assignment), double difference (or fixed effects), and single difference (or multivariate regression) [25]. I added to this a fifth level, between the first and second, for cluster randomized controlled trial. I calculated a simple score by assigning points 1 through 5 for identification strategy level and then adding a point each for concerns assessed, as described in Section A.1.4. Based on the clustering of the scores, I grouped the studies into strong, good, and weak confidence in the findings as discussed further in Section 3.

### 2.6. Analysis

For the primary studies, I report study characteristics and confidence-in-findings assessments for all 29 studies. To address risk of bias in the synthesis, I present and discuss the key findings separately according to the three categories of confidence in findings. To examine patterns in the findings for the strong- and good-confidence studies, I identify the level of the intervention as individual, community, or both. 

The purpose of the rapid evidence review is to provide useful information to my colleagues and other practitioners for immediate program design decisions, not to prove or disprove the effectiveness of an approach. Thus, I did not conduct meta-analysis. The heterogeneity of interventions and outcomes also precludes meaningful quantitative synthesis.

### 2.7. Inclusion of Systematic Reviews

The methods and results for the search and screening for systematic reviews is reported in Appendix C. As explained further below, none of the reviews provide synthesized evidence specific to meta-norms interventions’ effects on behavior change.

### 2.8. Departures from the Protocol

The primary departure from the registered protocol [20] is that the bulk of the review was conducted by a single reviewer instead of the two reviewers named in the protocol. The second reviewer, an intern, left the project earlier than expected. This unexpected change in staffing partly explains the delay in the completion of the review, which was shared with colleagues ten weeks after the protocol was registered instead of six. However, the reliance on a single reviewer also ensured consistency in screening, appraisal, and data extraction. Other departures from the protocol are reported in Section A.1.5.

## 3. Results

### 3.1. Literature Search

Figure 2 presents the PRISMA flow diagram for this review. After de-duplication, 1307 title and abstract records were screened, resulting in 216 records passing to full-text screening. A total of 89 full texts met the inclusion criteria. Coding for the scope of norms intervention—direct or meta—produced a set of 7 included systematic reviews and 29 included primary studies. The final set of 28 primary studies includes 1 substitution article [26], which reports the primary trial results related to an included study [27] that only reports secondary data analysis. The secondary analysis explains that there was a norms aspect to the intervention but does not include the primary findings from the evaluation.

### 3.2. Characteristics of Included Primary Studies

Figure 3 displays the heat map of the study locations by country. The majority of the studies come from Africa (21 out of 29), and there is roughly equal distribution (10, 9, and 10) of studies across low-income, lower-middle-income, and upper-middle-income countries. 

Table 1 presents the main features of each of the primary studies. All the documents report studies of distinct programs, except for two that evaluate that SASA! program [28,29]. The two documents for the SASA! evaluation, a journal article and an evaluation report, focus on different sets of outcomes. All but 2 of the 29 studies fall within the health or DRG sectors as the primary sector (not in table). Of the two exceptions, one tests a social marketing project to alter sustainable consumption behavior [30] (coded as environment and climate change), and the other studies an economic empowerment program to change education and employment behavior [31] (coded as economic growth). The majority of the studies concern HIV prevention [26,32,33,34,35,36,37], gender-based or intimate partner violence [38,39,40,41], or a combination of the two [28,29,42,43,44,45]. 

The DRG studies that are not about interpersonal violence include studies of: a community-based skills development program to delay child marriage [46]; a community development fund to increase pro-social behavior [47]; a school-based gender attitudes curriculum to improve boys’ behavior [48]; and a community-based program to encourage advocacy of gender equality [49]. The health studies that are not about HIV or interpersonal violence include: a community-based program to improve maternal health seeking [50]; a couples intervention to improve maternal health seeking and other behaviors [51]; a school-based program to reduce bullying [52]; a school-based sex education program to encourage safe sex behavior [53]; and community workshops to promote increased use of modern contraception [54]. The duration of interventions ranges from 1 month or less to 50 months. Half the interventions were shorter than 1 year; a third lasted between 18 months and 3 years; and the rest lasted 4 or more years.

**Table 1 ijerph-19-07312-t001:** Features of included primary studies.

Study ID	Title	Country	Program	Norms	Duration Months	Participant Age/Sex	Behavioral Outcome
Amin 2018 [46]	Skills-building programs to reduce child marriage in Bangladesh: A randomized controlled trial	Bangladesh	Bangladeshi Association for Life Skills, Income, and Knowledge for Adolescents (BALIKA)	Gender rights	18	Adolescents/Female	Personal
Avdeenko 2014 [47]	International interventions to build social capital: Evidence from a field experiment in Sudan	Sudan	Community Development Fund	Community pro-sociality	50	Adults/Together	Pro-social
Banerjee 2019 [32]	The entertaining way to behavioral change: Fighting HIV with MTV	Nigeria	MTV Shuga	Sexual health and HIV prevention	1	Youth/Together	Personal
Christofides 2020 [38]	Effectiveness of a multi-level intervention to reduce men’s perpetration of intimate partner violence: A cluster randomised controlled trial	South Africa	Sonke Community Health Action for Norms and Gender Equality (CHANGE)	Gender power, gender-based violence, sexuality	18	Adults/Male	Continuation
Cowan 2010 [33]	The Regai Szive Shiri project: Results of a ramonized trial of an HIV prevention intervention for youth	Zimbabwe	Regai Dzive Shiri	Sexual and reproductive health	48	Youth/Both	Personal
Dhar 2018 [48]	Reshaping adolescents’ gender attitudes: evidence from a school-based experiment in India	India	Breakthrough classes	Gender equality	30	Adolescents/Together	Continuation
Dougherty 2018 [50]	A mixed-methods evaluation of a community-based behavior-change program to improve maternal health outcomes in the upper west region of Ghana	Ghana	Community Benefits Health	Maternal health and breastfeeding	24	Adults/Female	Help seeking
Doyle 2018 [51]	Gender-transformative Bandebereho couples’ intervention to promote male engagement in reproductive and maternal health and violence prevention in Rwanda: Findings from a randomized controlled trial	Rwanda	Bandebereho	Gender equality, fatherhood, maternal health, intimate partner violence	5	Adults/Both	Help seeking
Figueroa 2016 [34]	Effectiveness of community dialogue in changing gender and sexual norms for HIV prevention: Evaluation of the Tchova Tchova program in Mozambique	Mozambique	Tchova Tchova	Gender equality, sexuality, HIV prevention	18	Adults/Together	Personal
Gottert 2020 [55]	Gaining traction: Promising shifts in gender norms and intimate partner violence in the context of a community-based HIV prevention trial in South Africa	South Africa	Tsima	Gender equality, relationship violence, sexual health	36	Adults/Male	Continuation
Kalichman 2013 [35]	Randomized community-level HIV prevention intervention trial for men who drink in South African alcohol-serving venues	South Africa	Alcohol–HIV risk reduction workshops	Safe sex, interpersonal communication	3	Adults/Male	Personal
Kraft 2012 [36]	Effects of the Gama Cuulu radio serial drama on HIV-related behavior change in Zambia	Zambia	Gama Cuulu Radio Serial Drama	Sexual health, sexual cleansing, child sexual abuse, HIV prevention	24	Adults/Together	Personal
Kyegombe 2014 [28]	The impact of SASA!, a community mobilization intervention, on reported HIV-related risk behaviours and relationship dynamics in Kampala, Uganda	Uganda	SASA!	Gender equality and power, gender-based violence	48	Adults/Both	Personal
Lundgren 2018 [49]	Does it take a village? Fostering gender equity among early adolescents in Nepal	Nepal	Choices, Voices, and Promises	Gender equity	3	Adolescents/Together	Personal
Maman 2020 [42]	Results from a cluster-randomized trial to evaluate a microfinance and peer health leadership intervention to prevent HIV and intimate partner violence among social networks of Tanzanian Men	Tanzania	Microfinance and peer health program for social networks of men	Gender-based violence and power, safe sex	30	Adults/Male	Continuation
Miller 2014 [39]	Evaluation of a gender-based violence prevention program for student athletes in Mumbai, India	India	Parivartan (Coaching Boys Into Men)	Gender equity, gender-based violence, abuse	4	Adolescents/Male	Personal
Naidoo 2016 [52]	Verbal bullying changes among students following an educational intervention using the integrated model for behavior change	South Africa	School-based education program	Gender-based violence, bullying	5	Adolescents/Together	Pro-social
Pettifor 2018 [43]	Community mobilization to modify harmful gender norms and reduce HIV risk: results from a community cluster randomized trial in South Africa	South Africa	Sonke One Man Can community mobilization	Sexual health, gender equality and power, gender-based violence, social cohesion	24	Youth/Together	Personal
Pulerwitz 2015 [40]	Changing gender norms and reducing intimate partner violence: Results from a quasi-experimental intervention study with young men in Ethiopia	Ethiopia	Male Norms Initiative	Gender equality, sexual health, intimate partner violence	1	Youth/Male	Continuation
Rijsdijk 2011 [53]	The world starts with me: A multilevel evaluation of a comprehensive sex education programme targeting adolescents in Uganda	Uganda	World Starts With Me	Sexual and reproductive health and rights	6	Adolescents/Together	Personal
Schuler 2015 [54]	Interactive workshops to promote gender equity and family planning in rural communities of Guatemala: Results of a community randomized study	Guatemala	C-Change couples workshops	Gender equality, sexual and reproductive health	1	Adults/Together	Continuation
Sharma 2020 [44]	Effectiveness of a culturally appropriate intervention to prevent intimate partner violence and HIV transmission among men, women and couples in rural Ethiopia: Findings from a cluster-randomized controlled trial	Ethiopia	Unite for a Better Life	Gender equality and power, sexuality, violence	2	Adults/Both	Continuation
Sosa-Rubi 2017 [41]	True Love: Effectiveness of a school-based program to reduce dating violence among adolescents in Mexico City	Mexico	True Love	Gender equality, dating violence, sexual rights	4	Adolescents/Both	Personal
Stark 2018 [31]	Effects of a social empowerment intervention on economic vulnerability for adolescent refugee girls in Ethiopia	Ethiopia	Creating Opportunities through Mentoring, Parental Involvement and Safe Spaces (COMPASS)	Gender power, reproductive health	10	Adolescents/Female	Personal
Thato 2013 [37]	A brief, peer-led HIV prevention program for college students in Bangkok, Thailand	Thailand	Brief, Peer-Led HIV Prevention Program	Sexual health, HIV prevention	1	Youth/Together	Personal
Vantamay 2019 [30]	“3S Project”: A community-based social marketing campaign for promoting sustainable consumption behavior among youth	Thailand	3S Project	Sustainable consumption	3	Youth/Together	Continuation
Wagman 2015 [45]	Effectiveness of an integrated intimate partner violence and HIV prevention intervention in Rakai, Uganda: Analysis of an intervention in an existing cluster randomised cohort	Uganda	Safe Homes and Respect for Everyone Project (SHARE)	Gender rights, intimate partner violence	4	Adults/Both	Continuation
Watts 2015 [29]	The SASA! study: A cluster randomised triral to assess the impact of a violence and HIV prevention programme in Kampala, Uganda	Uganda	SASA!	Gender equality and power, gender-based violence	32	Adults/Together	Continuation
Wechsberg 2016 [26]	The male factor: Outcomes from a cluster randomized field experiment with a couples-based HIV prevention intervention in a South African township	South Africa	Couples Health Coop, Women’s/Men’s Health Coop	Gender equality, sexuality, intimate partner violence, communication	1	Adults/Both	Personal

Of the eight studies conducted outside of Africa, four come from South Asia [39,46,48,49], two from Thailand [30,37], and one each from Guatemala [54] and Mexico [41]. The programs in South Asia and Latin America are all gender equality and gender-based violence programs. From Thailand, one study evaluates an HIV prevention program and the other a sustainable consumption project. 

Half of the studies measure outcomes for adults, six for youth, eight for adolescents, and none for children. Twelve studies measure outcomes for males and females combined, eight for males separately and females separately, three for only females, and six for only males. Although there were six possible coding categories for behavioral outcomes, almost all of the studies measure personal (freestanding) behaviors or behavioral continuation, with some fuzziness between these two categories, as discussed above. Two studies measure help seeking [50,51], and two studies characterize the behavioral outcomes as pro-social [47,52]. None of the studies measure service provider or policy- and decision-making behavioral outcomes. 

Data coded but not reported in the table show that only one study presents a cost-effectiveness analysis [29], and two others present limited cost information [33,36]. A recent report may assist future evaluators to provide costing information [56].

### 3.3. Confidence in Findings

Table 2 displays the results of the confidence-in-findings assessment according to the three groupings: strong confidence, good confidence, and weak confidence. Only one study is a randomized controlled trial (RCT); however, several are cluster randomized controlled trials (CRCT). It is not surprising to see the prevalence of cluster assignment when the interventions are targeting meta-norms, which often involve community-level activities. The 15 studies assessed as having strong confidence in the findings use random assignment and have one or fewer concerns. There are no studies that use as-if random assignment, so the step down to good confidence in the findings indicates random assignment studies with at least two concerns or studies that use double difference (DD) as the identification strategy and have no concerns. There are seven studies in this category. The seven studies assessed as having weak confidence in the findings all use single-difference (SD) comparisons for identification, that is, they do not control for any selection on unobservable variables, except for one DD study that also has one concern. Table 2 also lists the sample size used to estimate the primary behavioral outcome effect, but sample size was not a factor in the appraisal.

### 3.4. Estimated Effects on Behavior Change

Table 3, Table 4 and Table 5 present the main findings for the behavioral outcomes of the studies with strong, good, and weak confidence in the findings, respectively. The findings are mixed. Looking first at those appraised as strong and good (i.e., those with a design that controls for at least some unobservable characteristics), many programs did not demonstrate an effect on behavioral outcomes; for example, the Sonke CHANGE program in South Africa [38] and the related Tsima intervention implemented by Sonke [55] showed no effect on men’s perpetration of IPV, and the Regai Dzive Shiri program in Zimbabwe showed no effect on sexual or pregnancy prevention behavior [33]. Some programs demonstrated moderate or suggestive effects; for example, contraceptive use improved but not statistically significantly as an outcome of the C-Change couples’ workshops in Guatemala [54], and COMPASS in Ethiopia produced moderate improvements in schooling and transactional sex [31]. A few demonstrated large and statistically significant effects, such as the Bandebereho program in Rwanda, which produced substantial improvements in outcomes, including women’s attendance and men’s accompaniment at ante-natal clinics [51], and the alcohol–HIV risk reduction workshops in South Africa that caused men to increase condom use and engage in more conversations about HIV/AIDS [35]. 

While the presence of so many studies without positive findings may be an encouraging sign that there is no publication bias, many of the studies that did not detect changes in behavior did detect positive changes in other outcomes, such as attitudes and perceived social norms. Therefore, there may be additional evaluations that did not detect behavior changes or other changes that have not been published.

Findings are similarly mixed for the studies with weak confidence in the findings, shown in Table 5. The SHARE program in Uganda demonstrated detectable behavior changes in the disclosure of HIV status but did not demonstrate an effect on male-reported IPV perpetration [45]. The Parivartan program in India measured reduced bullying behaviors but only marginally statistically significantly [39]. The Tchova Tchova program in Mozambique increased the likelihood that respondents talked with their partners about HIV [34].

### 3.5. Analysis of Program Design

There is great heterogeneity in the activities included in the different programs, with most programs incorporating a mix of activities. Looking at the various sets of activities, though, we can see that some programs operate primarily at the community level, some at the individual level, and some at both. An example of a community-level program is the Community Development Fund in Sudan, which included community construction projects, social mobilizers, and community scorecards among other activities [47]. An example of an individual-level program is the use of Breakthrough classes in schools in India, which covered a curriculum of topics with students over time [48]. An example that combines the two levels is the True Love program in Mexico, which combined classroom-based workshops with community engagement activities [41]. I categorized the programs evaluated by the included studies in this review into the three groups—community, individual, or both—based on the descriptions of the activities in the studies. 

There is very limited evidence of positive behavioral outcomes from community-level programs from the studies with strong confidence in the findings. Avdeenko and Gilligan (2015) explain that, while the respondents in the community development fund communities in Sudan self-reported greater civic participation behavior than those in the control communities, when the authors examined participants’ behavior using a game, there were no differences [47]. The only community-level program among the good-confidence studies is the SASA! program in Uganda, which did produce positive outcomes for HIV prevention but did not measure direct behavioral outcomes for IPV [28,29]. The two studies of community-level programs among those with weak confidence did measure positive changes, one on maternal health behaviors [50] and the other on HIV communication behaviors [34]. These programs implemented different sets of activities, but perhaps of note, they had longer durations—18 to 48 months.

The most convincing positive findings among the strong and good evaluations of programs with activities at both the community level and individual level are from the alcohol–HIV risk reduction workshops in South Africa [35]. The men who participated in these workshops and community activities reported improvements in some, but not all, HIV risk reduction behaviors compared to the control. The True Love evaluation compares a combined community and individual intervention to only a community intervention and finds that the combined intervention reduces the perpetration of psychological violence relative to the community intervention alone [41]. Among the weak evaluations, the studies of programs implemented at both the community and individual level found limited positive outcomes.

There are two multi-arm studies that compare combined interventions to single interventions—one compares to community only and the other to individual only. The Pulerwitz, et al. (2015) [40] study, appraised as having good confidence in the findings, tests two variants against a control, one with community engagement activities only and one with community engagement and interactive group education for men. In this case, both elements are supposed to be norms-shifting, but the study offers a comparison of only working at the community level vs. working at both the community and the individual level. Both arms produced reductions in the perpetration of violence toward partners, although the community-engagement-alone arm has better statistical support. However, the combined intervention showed better results for other (non-behavioral) outcomes. The authors do not draw any conclusions about whether the combined activities are better than the community engagement intervention on its own.

The Lundgren, et al. (2018) [49] study, appraised to have weak confidence in the findings, compares an individual-level intervention (Choices) with that intervention combined with a family-level intervention (Voices) and a community-level intervention (Promises). All three are intended to change norms, so like the Pulerwitz, et al. study, it does not compare a “with norms approach” to a “without norms approach” but rather, individual-level approach to combined community, family, and individual. The study does suggest that the combined intervention had a greater effect on one of the measured behaviors. 

Among the individual-level programs, which account for the plurality of programs among the three appraisal categories, the findings are also mixed, with some programs demonstrating positive behavioral outcomes, for example, the classroom curriculum implemented by Breakthrough in India [48], and others not, for example, the microfinance and peer health leadership intervention in Tanzania [42]. No patterns emerge when looking at the features of the programs, including participants, duration, norms, and interventions. 

### 3.6. Systematic Reviews

As reported in Figure 2, the search, screening, and coding for meta-norms approaches identified seven systematic reviews. One explanation for the small number of systematic reviews screened is that the selection criteria (see Appendix A) required that the review report the country for each included study, include studies from LMICs, and present the findings to allow for separate synthesis of evidence for LMICs. An abbreviated ROBIS [57] risk-of-bias assessment categorized no reviews as having a low risk of bias; three as having an unclear risk of bias; and the other four as having a high risk of bias. Appendix C includes summaries of the three reviews with an unclear risk of bias, but while all three reviews included one or more study of interventions that had meta-norms-based approaches and include one or more study that estimated effects on measured behavior changes, the syntheses do not isolate the evidence for meta-norms-based approaches on behavior-change outcomes.

## 4. Discussion

Taken together, the studies included in this review reveal a growing evidence base for programs using social-norms approaches targeting meta-norms in low- and middle-income countries. No two interventions are the same, however, and the findings across studies are mixed with some strong positive effects and many marginal or null effects. From the evidence in this review, it is clear that meta-norms approaches can be effective to change behavior, but they are not always so. Put differently, incorporating meta-norms into programs is not sufficient to change behavior, as evidenced by the prevalence of studies in this review that show no effects on behavior.

For program design, it is also important to know whether norms shifting is necessary to produce behavior changes. There are many behavior theories applied to programming that do not involve norms. A primary example is Bandura’s social cognitive theory [58], which focuses on how people learn behaviors by observing others. On the other hand, prevailing social norms are considered key factors for some behaviors, for example, IPV, suggesting that these behaviors cannot be changed without shifting social norms. One way to explore this question is with multi-arm studies that compare with norms to without norms. This review only identified one such study.

The Amin, et al. (2018) [46] study is appraised as having a strong confidence in the findings. They implemented a four-arm CRCT in Bangladesh, where one arm was an educational support intervention, one was a gender rights awareness intervention, one was livelihoods training, and the fourth was the control. These were all community-based interventions, in the sense that the capacity development was delivered through community centers, but these were not community engagement interventions. The activities were delivered directly to the participating girls. Gender rights awareness is the most clearly a norms-shifting approach of the three. The primary outcome was child marriage. The finding is that all three interventions reduced child marriage without much measurable difference between them. Therefore, in this case, building girls’ skills in any of these three ways made a difference, and the gender rights awareness approach was not necessary.

Even if a norms approach is not necessary for a program to produce detectable changes in behaviors over a fixed period of time, it is likely still true that shifting social norms is necessary to produce sustainable, community-wide changes in behaviors such as IPV perpetration, for which social norms play a big role. The studies in this review are program evaluations, so what they show is which programs were able to achieve detectable changes in respondents’ behaviors over the evaluation period, which can inform the design of future programs that need to demonstrate the same.

The key findings reported in Table 3, Table 4 and Table 5 point to the studies for which positive behavioral outcomes can be detected. The heterogeneity in interventions, norms, and outcomes makes it impossible to draw conclusions about what works that can be applied across contexts. Program designers can use the evidence in this review by selecting the studies most relevant to the context, norms, participants, and outcomes they seek to address.

As reported in Table 1, the programs reviewed here targeted a variety of norms. The evaluations do not use a standardized framework for labeling the norms and provide limited information about the specific norms messages delivered. The information reported in Table 1 is thus descriptive without intending to establish categories across studies. Just as meta-norms influence multiple behaviors, there are behaviors that may be influenced by multiple norms. For example, IPV perpetration may be influenced by norms around violence and criminality and also by norms around family responsibility. Similarly, HIV risk reduction may be influenced by norms around personal responsibility and self-care and also by norms around responsibility to others. In such cases, for program design, we want to know if one norm is more effective than another at changing behavior. Unfortunately, the studies in this review do not allow us to answer these questions. There are no multi-arm studies implementing the same activities but with different norms messages. In addition, there is not enough information about norms messages or consistency in approaches across studies that we can attribute the success of one program compared to another as explained by the specific norms targeted.

In spite of the growing evidence base, there are challenges for implementers and program designers to incorporate the latest learning on social norms. One challenge is simply getting the concepts and terminology straight. First, we can think of two kinds of norms approaches. The original social norms theory for behavior change is not about changing norms but about changing individuals’ knowledge or perceptions of what the existing social norms are in order to change their behavior, for example, the college drinking and utility bills experiments. These interventions often use direct norms, that is, the norm directly associated with the behavior. 

In designing and implementing programs for development funders, we are more often tasked with changing, or shifting, norms in order to change behavior. The programs that address meta-norms, reviewed here, are generally of the latter type—trying to change norms. The interventions that use norms are generally quite different than the interventions that change norms, so evidence from one type should not be used to design the other.

Second, within the numerous interventions that seek to change norms, there is a fuzzy line between actually addressing social norms (what others do and believe) and simply addressing attitudes (what each individual thinks), which eventually changes norms through aggregation. This fuzziness is one reason that measuring changes in norms is complicated—people may misperceive or misattribute their own attitude change to a change in social norms. Cislaghi and Shakya (2018) argue that “the norms as attitudes school is not helpful for practitioners dealing with pluralistic ignorance…” [5] (p. 40). Many of the studies excluded from the review measure norms as an outcome even though the interventions addressed only knowledge and attitudes.

### Limitations

The limitations to this review mostly arise from the rapid-review design decisions. The review is biased toward more recent studies, although the systematic reviews include numerous studies published before 2010. The narrowed search strings mean that some eligible studies were certainly missed. The public health rapid reviews reported in Affengruber, et al. (2020) [15] included fewer than half the included studies in the reference review. Those rapid reviews relied on an electronic search of a single index and reference list checking, and the missed studies were all non-indexed studies. The methodology used here improves on that by electronically searching multiple indexes to yield a much larger number of abstracts to screen and by searching websites to find grey and non-indexed studies.

The English-only restriction is a notable limitation, especially when reviewing studies conducted in low- and middle-income countries. The review is missing, for example, research conducted in Francophone Africa and published in French. We balanced this limitation against the demand to conduct a rapid review using available resources.

One reviewer for all screening and coding improves the consistency relative to a rapid-review design where multiple reviewers screen separate studies. However, bias and mistakes in screening and coding are more likely when there is only one reviewer per record instead of screening and coding in duplicate [15]. The review question covers research published in both health and social science outlets. Based on my experience with this and an earlier review [59], I suspect that the search missed more eligible social science articles than health articles. One indication is that the number of hits from EconLit was relatively low (55 hits). One explanation for this bias is that social science abstracts do not follow a consistent template and often leave out key details, and some social science journals do not use abstracts at all. The definition of what is a norms-based approach is unavoidably imprecise, meaning some of the screening decisions required more of a judgment call than others.

## 5. Conclusions

In the last decade, dozens of behavior-change interventions using norms-based approaches have been rigorously evaluated. The majority of these address a direct norm tied to a specific behavior; however, 29 studies evaluate behavior-change outcomes for 28 different programs that address meta-norms, such as gender rights, reproductive health, sexuality, and IPV. In addition, seven systematic reviews published since 2010 examine behavioral outcomes from a wide variety of interventions, some of which include meta-norms-based approaches. Taken together, the evidence shows there are some effective programs that include norms-based approaches, but addressing meta-norms does not guarantee that a program will produce individual behavior changes.

The scope of interventions eligible for this review was broad, allowing for all sectors, themes, and activities and allowing for any kind of meta-norm. As such, the included programs are highly heterogeneous. While this complicates evidence synthesis, it should reflect that the programs are being designed with each context and relevant social norms in view.

This review finds that the evidence base for behavior-change interventions using meta-norms-based approaches primarily covers HIV prevention, gender-based or intimate partner violence, or a combination of the two. Program designers in those fields will find the most relevant evidence here, especially for countries in Africa. Program designers for meta-norms approaches in other fields can also use this review to look for studies relevant to their evidence needs. All the studies from South Asia and Latin American evaluate programs addressing gender equality and gender-based violence. 

I recommend that program designers seeking to shift meta-norms consider including community-level activities in their programs, and that programs implemented only at the community level be implemented over a longer period of time before expecting to detect behavior changes. Because the effectiveness of meta-norms approaches varies by context and program, program designers should conduct both formative research to inform their program design and formative evaluation to test for feasibility and acceptability of the proposed intervention prior to starting a full implementation. Some studies reviewed here, for example, Gottert, et al. (2020) [55], find positive outcomes for both the program and comparison groups, so new programs should also be subject to impact evaluation (i.e., evaluation against the counterfactual) before going to scale to ensure that positive outcomes can be attributed to the program.

In terms of future research, this review reminds us that it is necessary to measure behavioral outcomes when evaluating programs intended to change behavior. Many studies were excluded because they did not actually measure behavioral outcomes, and in the included studies, we see that positive outcomes for knowledge, attitudes, perceptions, and intent do not always translate into behavior change. Researchers should also pay more attention to how they are analyzing and interpreting findings when they measure a large number of outcomes. The multiple comparisons concerns caused a handful of the included studies to be downgraded during critical appraisal. It is difficult for program designers to use evidence to predict the effects of their planned program when there are only a few positive findings among a large number of measured outcomes. Finally, future research will also be more useful to decision makers and accessible for review if evaluators more carefully document the norms targeted and which activities are designed specifically to shift norms, especially when evaluating complex programs.

## Figures and Tables

**Figure 1 ijerph-19-07312-f001:**
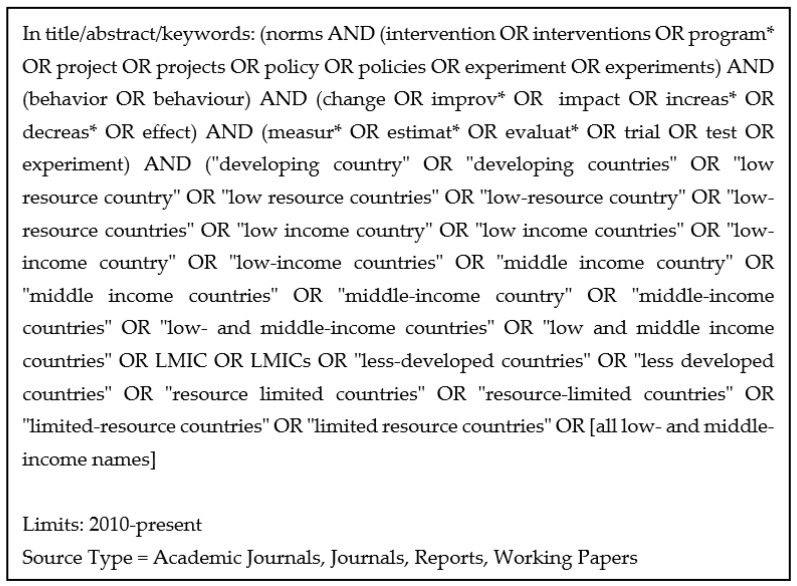
Search strategy for EBSCO platform.

**Figure 2 ijerph-19-07312-f002:**
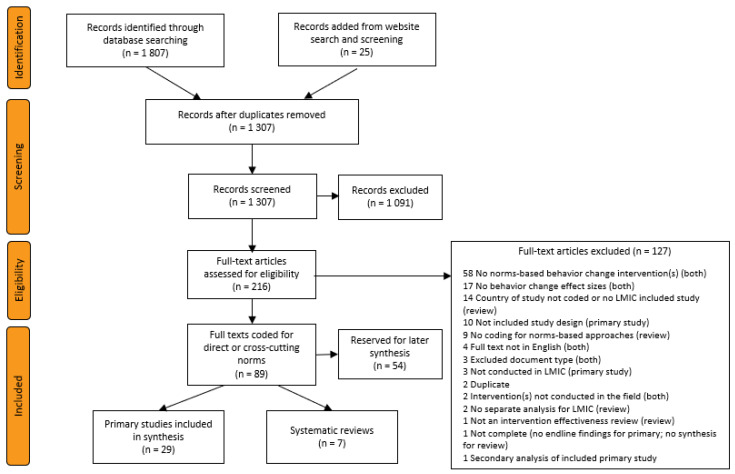
PRISMA flow diagram of search and screening results.

**Figure 3 ijerph-19-07312-f003:**
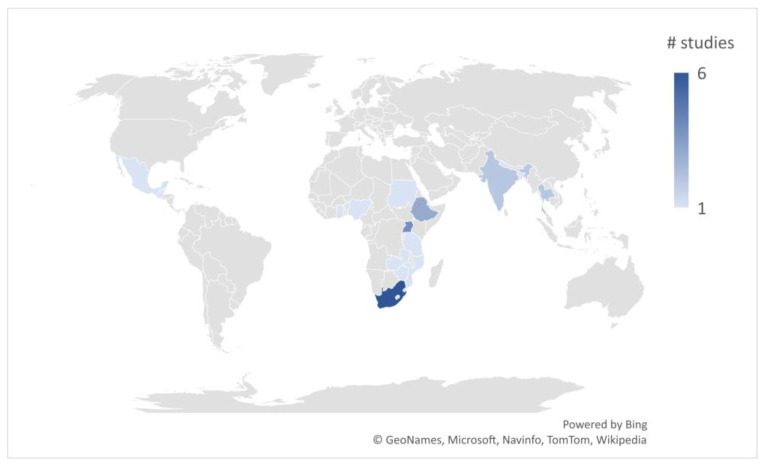
Heat map of primary studies.

**Table 2 ijerph-19-07312-t002:** Confidence-in-findings appraisal and sample size for behavioral outcome of primary studies.

Study ID	Identification Strategy	Number of Concerns	Concern Notes	Sample Size
**Strong confidence**				
Amin 2018 [46]	CRCT	0	None	5309
Avdeenko (2015) [47]	CRCT	0	None	576
Banerjee 2019 [32]	CRCT	1	Working paper	3070
Christofides 2020 [38]	CRCT	0	None	1508
Cowan 2010 [33]	CRCT	1	High attrition from outmigration	2079
Dhar 2018 [48]	CRCT	1	Working paper	13,987
Doyle 2018 [51]	RCT	1	Multiple comparisons	1123
Gottert 2020 [55]	CRCT	0	None	915
Kalichman 2013 [35]	CRCT	1	Multiple comparisons	984
Maman 2020 [42]	CRCT	0	None	1029
Pettifor 2018 [43]	CRCT	1	Repeated cross-section used to measure individual changes	2356
Schuler 2015 [54]	CRCT	1	High lost to follow-up	294
Sharma 2020 [44]	CRCT	0	None	5125
Stark 2018 [31]	CRCT	0	None	881
Wechsberg 2016 [26]	CRCT	0	None	533
**Good confidence**				
Kyegombe 2014 [28]	CRCT	2	Challenges to implementation, social desirability bias related to intervention, multiple comparisons	2530
Naidoo 2016 [52]	CRCT	2	High attrition, behavioral metric unclear, no correction for clusters	434
Pulerwitz 2015 [40]	DD	0	None	483
Sosa-Rubi 2017 [41]	DD	0	None	885
Thato 2013 [37]	DD	0	None	30
Vantamay 2019 [30]	DD	0	None	80
Watts 2015 [29]	CRCT	2	Institutional report, challenges to implementation	1459
**Weak confidence**				
Dougherty 2018 [50]	SD	3	Confusion over the implementation of the intervention incentives, evidence of spillover, no correction for clusters	1050
Figueroa 2016 [34]	SD	1	Multiple comparisons, weak indexes, social desirability bias related to intervention	915
Kraft 2012 [36]	SD	1	Implementation challenges	3624
Lundgren 2018 [49]	DD	2	Shortened intervention period, multiple comparisons, no true control	1200
Miller 2014 [39]	DD	2	Evidence of implementation fidelity problems, high lost to follow-up	309
Rijsdijk 2011 [53]	DD	2	Intervention not implemented with fidelity in all sites, multiple outcomes	1519
Wagman 2015 [45]	DD	2	Attrition, multiple comparisons	2953

**Table 3 ijerph-19-07312-t003:** Main findings for behavioral outcomes from primary studies assessed as having strong confidence in the findings.

Study ID and Program	Intervention Level	Intervention Activities	Key Findings on Behavioral Outcomes
Amin 2018 [46] BALIKA	Individual	Training centers for girls offering education assistance, gender rights awareness training, or livelihoods skills; mentors	“The average program effect in the community was significant with adjusted hazard ratio…0.72…for gender…relative to those living in control arm villages.” p. 29
Avdeenko 2015 [47] Community Development Fund	Community	Community construction projects, capacity building for project management and community participation, social mobilizers, community scorecards	“Respondents in treated communities self-report greater civic participation than do respondents in control communities.” p. 441 However, “Subjects in treated communities did not behave more prosocially in any of the laboratory activities than did subjects from control communities.” p. 438
Banerjee 2019 [32] MTV Shuga	Community	Edutainment, television serial drama	The study estimates that MTV Shuga led to reduction in risky sexual behavior as measured by an index, but the effect is not statistically significant. p. 23
Christofides 2020 [38] Sonke CHANGE	Both	Door-to-door discussions, community action teams, workshops	“We found that the intervention did not significantly affect any of the primary or secondary outcomes. There was no effect on men’s past year use of physical or sexual IPV or a reduction in severe IPV.” p. 10
Cowan 2010 [33] Regai Dzive Shiri	Both	Youth program delivered by professional peer educators, community-based program for knowledge and support, training for clinic workers	“There was no effect of the intervention on reported sexual behavior, reported clinic use or reported use of pregnancy prevention in men or women in intervention communities.” p. 2549
Dhar 2018 [48] Breakthrough classes	Individual	School-based classroom discussions	“...self-reported behaviors influenced by gender attitudes...became more aligned with gender-progressive norms...” p. 15
Doyle 2018 [51] Bandebereho	Individual	Small group sessions with curricula for couples and men and women separately	“The Bendebereho intervention led to substantial improvements in multiple reported outcomes, including … women’s ANC attendance, men’s accompaniment at ANC, modern contraceptive use, and partner support during pregnancy.” p. 12
Gottert 2020 [55] Tsima (by Sonke)	Both	Workshops for men and women, street theater and community activities, support groups, local leadership engagement	“Younger men in both intervention and control communities reported reductions in IPV perpetration, leading to a null intervention effect.” p. 14
Kalichman 2013 [35] Alcohol-HIV risk reduction workshops	Both	Individual-level workshops, role playing, community-level activation	“However, there was a significant effect of the intervention on proportion of intercourse occasions protected by condoms; the experimental HIV prevention groups demonstrated significantly greater use of condoms over the follow-up period. Also, men in the intervention condition engaged in more conversations within their communities about HIV/AIDS…” p. 836
Maman 2020 [42] Microfinance and peer health program	Individual	Business and entrepreneurship training, microfinance loans, investment groups, peer health leader training, peer health discussions	For men, “there were no differences in condition in STI prevalence, IPV perpetration, or sexual risk behaviors at the 30-month follow-up.” p. 1
Pettifor 2018 [43] Sonke One Man Can	Community	Community workshops, community action teams, community theater and discussion, leader engagement	“When examining secondary endpoints for the trial, we did not observe significant differences among men in intervention versus control communities over time regarding multiple sex partners in the past 12 months, condom use at last sex, perpetration or experience of intimate partner violence or hazardous drinking.” p. 6
Schuler 2015 [54] C-Change couples workshops	Individual	Interactive workshops for couples (together and separate), games, role playing	“Findings regarding contraceptive use were suggestive but not significant.”
Sharma 2020 [44] Unite for a Better Life	Individual	Participatory and skills-building sessions delivered as part of coffee ceremonies	“For the secondary outcomes, only the men’s UBL intervention significantly reduced male perpetration of past-year sexual IPV...and no intervention reduced perpetration of past-year physical IPV.”
Stark 2018 [31] COMPASS	Individual	Weekly sessions for girl groups with mentors in traditional huts, parental engagement	“...girls in the treatment arm had approximately equal odds compared to girls in the control arm of working for pay at the end line... We observe moderate trends in the hypothesized directions for schooling and transactional sexual exploitation...” p. S17
Wechsberg 2016 [26] Couples’, Women’s and Men’s Health Coops	Individual	Peer-led workshops with groups in community libraries, including role playing and action planning	“Men in the CHC arm were half as likely to report heavy drinking at 6-month follow-up as men in the comparison arm…The proportion of men reporting consistent condom use in the past 30 days increase in each intervention arm.” p. 313

**Table 4 ijerph-19-07312-t004:** Main findings on behavioral outcomes for primary studies assessed as having good confidence in the findings.

Study ID	Intervention Level	Intervention Activities	Key Findings on Behavioral Outcomes
Kyegombe 2014 [28] SASA!	Community	Community activists, training of professionals, door-to-door visits, community dramas and films, community conversations	“Among men, effect estimates in the hypothesized direction were observed for all HIV-risk behaviours and indicators of relationship dynamics, with results statistically significant at the 5% level for all but two outcomes.” p. 4
Naidoo 2016 [52] School-based education program	Individual	Weekly lessons in school with group discussion, role playing, video watching, and drawing	“Comparing the verbal bullying of other people in the intervention group versus control, from baseline to postintervention (*p* = 0.047) was significantly reduced.” p. 817
Pulerwitz 2015 [40] Male Norms Initiative	Both	Interactive group education, community mobilization and engagement, Engaging Boys and Men in Gender Transformation manual	“In our multivariate IPV analyses, however, only the finding of lower reported violence from the community engagement (CE-only) intervention remained marginally significant...” p. 136
Sosa-Rubi 2017 [41] True Love	Both	Training of school staff, schoolyard activities, classroom-based workshops, community engagement	“We found a 58% (*p* < 0.05) and 55% (*p* < 0.05) reduction in the prevalence of perpetrated and experienced psychological violence, respectively, among SCC, IL-1 males compared to males exposed only to the SCC component.” p. 804
Thato 2013 [37] Brief, Peer-led HIV Prevention Program	Individual	Group sessions taught by peer leader using video and discussion	The program “did not significantly increase AIDS/STIs preventative behaviors” p. 58
Vantamay 2019 [30] 3S Project	Individual	Social marketing campaign through classroom activities and projects	“The mean scores of the experimental group were higher than the mean scores of the control group significantly in all of the five indicators... including the sustainable consumption behavior.” p. 42
Watts 2015 [29] SASA!	Community	Community activists, training of professionals, door-to-door visits, community dramas and films, community conversations	“The findings also suggest that SASA! impacted significantly on the reported levels of sexual concurrency, with 27 per cent of partnered men in intervention communities reporting having had other sexual partners in the past year, compared to 45 per cent of men in the control communities…” p. ii

**Table 5 ijerph-19-07312-t005:** Main findings on behavioral outcomes for primary studies assessed as having weak confidence in the findings.

Study ID	Intervention Level	Intervention Activities	Key Findings on Behavioral Outcomes
Dougherty 2018 [50] Community Benefits Health	Community	Community-based incentives, health messaging through various media, peer educators and community health officers	“Across three of the six study outcomes, we found that women who were exposed to the intervention activities were significantly more likely to have practiced improved health outcomes.” p. 88
Figueroa 2016 [34] Tchova Tchova	Community	Facilitated community dialogues, radio magazine, HIV/AIDS prevention gender tool	“88% of intervention and 72% of control respondents in the sample said they talked with their partner about HIV and/or sexual behavior in the past 3 months.” p. 559
Kraft 2012 [36] Gama Cuulu Radio Serial Drama	Both	Radio serial drama, community drama, one-on-one education sessions, small-group discussions	“There were few statistically significant differences over time between the two provinces, and only one statistically significant difference on a behavioral outcome (i.e., condom use, last sex).” p. 941
Lundgren 2018 [49] Choices, Voices, and Promises	Both	Video series for families, group discussion, public poster display, community discussion	In the Choices, Voices, Promises sample, respondents were more likely to answer they have told their parents (guardian) that it is important for sisters/you to continue studying than in the Choices only sample.
Miller 2014 [39] Parivartan (Coaching Boys into Men)	Individual	Training of coaches, coach role models, coach-led discussions with players	“Fewer negative intervention behaviors (i.e., laughing and going along with peers’ abusive behaviors) were reported by intervention athletes at follow-up compared with comparison athletes, but this difference was only marginally significant.” p. 771
Rijsdijk 2011 [53] World Starts with Me	Individual	Computer-based lessons in schools with peer educators, games, assignments	“No significant effects were found for past performance behaviour [of condom use].” “No significant effects were found for past performance behaviour regarding avoiding and escaping risky situations…” p. 8
Wagman 2015 [45] SHARE	Both	Community-based mobilization through advocacy, training, learning materials, and special events plus one-on-one intervention with women seeking HCT	“SHARE has no effect on male-reported IPV perpetration.” p. e23 “SHARE was also associated with significant increases in disclosure of HIV status in men and women.” p. e30

## Data Availability

The data presented in this study are openly available in Open Science Framework at osf.io/qmg3k.

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
