# Peer review of "Some Interventions to Shift Meta-Norms Are Effective for Changing Behaviors in Low- and Middle-Income Countries: A Rapid Systematic Review"

_ijerph, 2022, doi:10.3390/ijerph19127312_

Round 1
Reviewer 1 Report
The article contains a review about the evidence for the effectiveness of interventions that use norms-based approaches to change behaviour.
The literature research was done in a very thorough way and with a large basis of primary studies and systematic reviews. All the details of the study method are described. Especially figure 2 is helpful for the understanding of the method.
The content of the study is very relevant and interesting to the readers.
However, I would suggest to improve two aspects of the article:
- The article is written in a lab-journal/diary sort of style, e.g. “The librarian conducted the electronic searches on 2 November 2020, and the second reviewer conducted the website searches between 2 and 6 November 2020.”, ” When the screening was complete, I searched for journal articles associated with..”, ” The librarian exported the search results from the index searches into Endnote”, ” The second reviewer conducted the website searches”, ” The second reviewer, an intern, left the project earlier than expected. This unexpected change in staffing partly explains the delay in the completion of the review”.
- I would propose to shorten the article (especially section 2) by around 20-30% and just leave the essential part for understanding the method.
- I’m also missing some sort of take-away message. In the conclusion the author writes that “Program designers can draw some general conclusions from this review, but can also use it to locate specific studies relevant to their evidence needs”. It is true that the tables are interesting in themselves, but it would be helpful to have a list of 5 to 10 insights or recommendations, why some of the programs were efficient and others not. Earlier in the article the author writes: “I would recommend that designers planning a program implemented only at the community level design the program to be implemented over a longer period of time before expecting to detect behavior changes.” It would be good to have more of this type of recommendations and also more prominent in the conclusion of the article.
Apart from these two suggestions I have no other recommendations for improvement of this article.
Author Response
I would like to thank the reviewer for helpful suggestions.
- I have cut 25% of the text from the methods and materials section and moved it to an appendix. The lab-journal format is common for a Cochrane-Campbell systematic review, but I agree it makes for hard reading. The data point for the date of the search is critical systematic review information, so I have left that in the main body. I also left of the description relevant the rapid review methods and results as one contribution of this review is the model of a rapid review.
- I have expanded the conclusions section of the manuscript to include more insights and recommendations from the research. I moved some points from other sections of the manuscript so that the insights and recommendations are consolidated in one place, and I have also added a few more points.
Reviewer 2 Report
This rapid review was conducted to inform the design of social and 10 behavior change programs in low- and middle-income countries in response time-sensitive demands and it examined the evidence for the effectiveness of interventions that use norms-based approaches to change behavior.
Nine indexes and eight websites were electronically searched for both systematic reviews and primary studies. Both abstracts and full texts were screened to include: documents published in 2010 and later; evaluating the effectiveness of programs that include norms-based approaches; measuring behavioral outcomes; and employing quantitative analysis of concurrent treatment and comparison groups.
The results of the review show that meta-norms based approaches can be part of effective programs but do not assure that programs will change behaviors. Program designers can draw some general conclusions from this review, but can also use it to locate specific studies relevant to their evidence needs.
The author argued that as compared to other rapid reviews that limit their searches to existing systematic reviews, some to primary studies, and some include both. The full rapid review was conducted for this study including both systematic reviews and primary studies as eligible studies.
Although there is general agreement that rapid review methodologies produce less reliable findings than full systematic reviews
Despite the inherent drawback of the rapid review, the author clearly highlights the importance of rapid reviews to specific evaluation of programs to serve the needs of decision makers, and to help meet those needs,
The context is relevant; the topic is extremely important as the conduct the full rapid review including both systematic reviews and primary studies as eligible studies can provide robust evidence like the systematic review for the evaluation of programs to serve the needs of decision makers, and to help meet those needs. Nevertheless, I have the following queries and suggestions:
Major comments
1. Address the issue of only English literature included since Francophone speaking countries in Africa have the highest FGM and several programs are taking place there (Senegal, Burkina Faso, Mali, Guinee etc….). This can enforce the idea that only English literature is accessible but for example Senegal a French speaking country has been running a very successful program with the work of Tostan hence the need to include French literature.
2. To generalize these findings, break down the findings by region or countries to see the coverage of the findings.
3. In the presence highly heterogeneous programs and mixed findings, how can the author justify the conclusion that “Taken together, the evidence shows that meta-norms based approaches can be part of effective programs but do not assure that programs 26 will change behaviors.” I think, prudence should be exercised here.
Author Response
I would like to thank the reviewer for useful comments and suggestions.
- I agree this is a notable limitation. I have noted in section 2.1 that the English language restriction is a limitation, and I have added a discussion of limitation in the limitations section (4.1.)
- I agree that the manuscript can benefit for more discussion of the regional distribution of the evidence base. Given the length of the manuscript and the size of the current tables, I decided not to add a table with the same information presented in a different way. However, I have added discussion of regions and countries to section 3.2 and to the conclusions.
- I understand that the word "can" might be confusing. Here I mean "is possible" rather than "does". As you suggest, I certainly do not want readers to come away with a stronger interpretation than I mean. I have addressed this concern by changing the title of the manuscript and by making clarifications in the conclusions.
Reviewer 3 Report
The paper submitted by the authors is interesting and presents valuable insights. However, in my opinion it is a report rather than a scientific article. It could be concluded that the conducted "rapid review" is, in fact, a study of a research problem based on the literature on the subject, if it was an introduction to one's own empirical research. In order for the paper to become scientific, it is necessary to formulate a specific research problem and clarify the research aim, which will be solved later on. This also needs to be supported by a relevant methodology, going beyond the compilation of the other authors’ findings. If the Authors do not find it advisable to conduct their own research, it is absolutely necessary to at least ground the considerations in scientific theories related to the subject matter. Therefore I ask the Authors to edit the text so that, in addition to its utilitarian values (which the submitted paper undoubtedly has), the paper brings new cognitive values beyond the literature review.
Author Response
I would like to thank the reviewer for this feedback.
This manuscript was submitted under the systematic review article type. The scientific method applied for this research is the standard Cochrane-Campbell approach to systematic reviews with adaptations to allow for rapid review, as noted in the manuscript. The systematic review approach to literature review is an empirical method, where existing studies constitute the observations, and systematic search and screening are conducted to collect an unbiased sample. Data are then coded and analyzed for each of the observations in the sample. As with many systematic reviews, especially covering heterogeneous interventions and outcomes, this review does not conduct statistical analysis of the extracted outcomes. The registered protocol for this systematic review, cited in the manuscript, presents the methods. The supplementary materials include a PRISMA checklist for the review.
In response to this feedback, I have more clearly stated the research question from the protocol in the objectives section (1.2) of the manuscript. I have also added more insights and recommendations to the conclusions section of the manuscript.
Round 2
Reviewer 1 Report
The two aspects I have mentioned in the first review (style & lenght of text; conclusion) have been improved. So, I would recommend to publish the article in the present form.
Reviewer 2 Report
My previous comments/suggestions have been satisfactory address and I am happy with the current version of the paper.
Reviewer 3 Report
The author took into account most of the reviewer's suggestions. Now the article has a research goal, references to the theory and improved conclusions. In my opinion, it is suitable for publication in the journal.